# Association between breastfeeding, host genetic factors, and calicivirus gastroenteritis in a Nicaraguan birth cohort

Nadja Alexandra Vielot[1‡]*, Ruthly François[2‡], Emilya Huseynova[3], Fredman González[4], Yaoska Reyes[4], Lester Gutierrez[4], Johan Nordgren[5], Christian Toval-Ruiz[4], Samuel Vilchez[4], Jan Vinjé[6], Sylvia Becker-Dreps[1,4‡], Filemon Bucardo[4‡]

1 Department of Family Medicine, University of North Carolina at Chapel Hill, Chapel Hill, North Carolina, United States of America, 2 School of Medicine, University of North Carolina at Chapel Hill, Chapel Hill, North Carolina, United States of America, 3 Department of Epidemiology, University of North Carolina at Chapel Hill, Chapel Hill, North Carolina, United States of America, 4 Department of Microbiology and Parasitology, National Autonomous University of Nicaragua–León, León, Nicaragua, 5 Department of Biomedical and Clinical Sciences, Linköping University, Linköping, Sweden, 6 Division of Viral Diseases, Centers for Disease Control and Prevention, Atlanta, Georgia, United States of America

‡ NAV and RF are co-first authors, SBD and FB are co-senior authors.
* nadjavielot@unc.edu

**Data Availability Statement:** All relevant data are within the paper and its Supporting Information files.

## Abstract

### Background

Norovirus and sapovirus are important causes of childhood acute gastroenteritis (AGE). Breastfeeding prevents AGE generally; however, it is unknown if breastfeeding prevents AGE caused specifically by norovirus and sapovirus.

### Methods

We investigated the association between breastfeeding and norovirus or sapovirus AGE episodes in a birth cohort. Weekly data on breastfeeding and AGE episodes were captured during the first year of life. Stools were collected from children with AGE and tested by RT-qPCR for norovirus and sapovirus. Time-dependent Cox models estimated associations between weekly breastfeeding and time to first norovirus or sapovirus AGE.

### Findings

From June 2017 to July 2018, 444 newborns were enrolled in the study. In the first year of life, 69 and 34 children experienced a norovirus and a sapovirus episode, respectively. Exclusive breastfeeding lasted a median of 2 weeks, and any breastfeeding lasted a median of 43 weeks. Breastfeeding in the last week did not prevent norovirus (HR: 1.09, 95% CI: 0.62, 1.92) or sapovirus (HR: 1.00, 95% CI: 0.82, 1.21) AGE in a given week, adjusting for household sanitation, consumption of high-risk foods, and mother's and child's histo-blood group phenotypes. Maternal secretor-positive phenotype was protective against norovirus AGE, whereas child's secretor-positive phenotype was a risk factor for norovirus AGE.

**Funding:** This study was supported by award R01AI127845 from the National Institute of Allergy and Infectious Diseases (NIAID). SBD is supported by K24AI141744 from NIAID; FG, YR, and LG are supported by an international research capacity-building award from the NIH-Fogarty International Center, D43TW010923. The funders had no role in study design, data collection and analysis, decision to publish, or preparation of the manuscript. https://www.niaid.nih.gov/ https://www.fic.nih.gov/ .

**Competing interests:** The authors have declared that no competing interests exist.

## Interpretation

Exclusive breastfeeding in this population was short-lived, and no conclusions could be drawn about its potential to prevent norovirus or sapovirus AGE. Non-exclusive breastfeeding did not prevent norovirus or sapovirus AGE in the first year of life. However, maternal secretor-positive phenotype was associated with a reduced hazard of norovirus AGE.

## Introduction

Acute gastroenteritis (AGE) is a common cause of morbidity and mortality worldwide, with the greatest burden among children in developing countries [1]. Human caliciviruses, which include norovirus and sapovirus, are a leading cause of AGE in both outbreak and endemic settings [2, 3]. Although people of all ages are susceptible to norovirus infections, the highest morbidity and mortality is among young children and the elderly [2]. Sapovirus is increasingly recognized as an important etiology of AGE among young children, with the prevalence ranging from 5–17% depending upon the country, age group studied, and laboratory methods used [4–7]. In the MAL-ED multi-site cohort study, sapovirus has been recognized as the second highest attributable incidence of AGE in children under 24 months of age [8].

The protective effect of breastfeeding against childhood AGE is well established [9]. However, few studies have investigated the association between breastfeeding and specific causes of AGE, such as from norovirus and sapovirus infections. The current evidence from the published literature shows mixed results for norovirus and sapovirus; some studies have shown that breastfeeding is associated with a decreased odds of infection, while others have shown no reduction [10–13]. Moreover, when breastfeeding was assessed in these studies, there were inconsistencies and lack of clarity in how breastfeeding frequency and exclusivity were treated as analytic variables.

Along with breastfeeding, child and mother's secretor phenotype affects individuals' susceptibility to enteric infections. More specifically, the mother's secretor phenotype can alter breastmilk composition [14, 15], and the child's secretor phenotype can increase or decrease the risk for diarrheal diseases of specific etiology, including norovirus genogroups [16, 17]. However, limited evidence exists on the association between detailed breastfeeding history, child's and mother's histo-blood group antigens (HBGAs) phenotypes, and the risk of norovirus or sapovirus AGE [18].

Our research investigated the protective association between breastfeeding and incidence of first norovirus and sapovirus AGE episodes during the first 12 months of life in a longitudinal birth cohort study (SAGE) in León, Nicaragua. The SAGE birth cohort employs a rigorous follow-up approach and a platform to study the weekly impact of breastfeeding on the incidence and severity of norovirus and sapovirus infections. A secondary goal of the study was to understand the combined impacts of child and maternal genetic factors on norovirus and sapovirus AGE risk, and whether these influence the benefits of breastfeeding on AGE prevention.

## Materials and methods

### Study design

The Sapovirus Associated GastroEnteritis (SAGE) study is a population-based birth cohort in Léon, Nicaragua that aims to investigate the natural history, immunity, and transmission

patterns of sapovirus and 13 other enteric pathogens associated with pediatric gastroenteritis. The target sample size to detect a three-fold risk of sapovirus infection with 90% power among children with an infected household member was 400. During the recruitment period between June 2017 and July 2018, 991 expected births were identified from exhaustive pregnancy registries maintained by public health centers, which were highly representative of the full population of León. A total of 742 mothers were contacted in their homes, of whom 585 were eligible to participate and 444 agreed to participate (recruitment rate: 76%). Newborns were enrolled between birth and 14 days of age and were followed weekly for up to 36 months. Ten fieldworkers underwent training in administering surveys and collecting biological samples, then conducted in-person household visits to collect baseline data on sociodemographic factors; weekly data on breastfeeding behavior, supplemental feeding, and incidence of AGE episodes; and monthly data on environmental and behavioral risk factors for AGE. Fieldworkers participated in biweekly meetings to review study protocols and update trainings.

## Measures

The main outcome was the first episode of AGE in which norovirus or sapovirus was detected in the stool. AGE surveillance occurred weekly, and an AGE episode was characterized by vomiting and/or diarrhea, defined as $\geq 3$ stools that were loose or looser than normal in a 24-hour period, including notable changes to the stool, such as presence of blood or excessive liquid. This analysis was restricted to the first detected episode of norovirus or sapovirus. Breastfeeding practice was ascertained using a weekly questionnaire administered to the mothers. Children for whom the answer to the question "Did you breastfeed your child yesterday?" was "Yes" were considered breastfed. Children were considered to be exclusively breastfed until the first week the mother answered "Yes" to the question "In the past week, has your child received any food, formula, or liquids, including water or tea or liquid in a bottle, in addition to breast milk?"; children were considered non-exclusively breastfed thereafter. Children were considered to have "any breastfeeding" if they received either exclusive or mixed breastfeeding at any point in the first 12 months of the study.

Severity of norovirus or sapovirus episodes was summarized on a scale of 0–15 based on the presence of the following symptoms and treatment-seeking behaviors: diarrhea lasting 1–2 days = 1 point, 3–4 days = 2 points, 5+ days = 3 points; vomiting lasting 1–2 days = 1 point, 3–4 days = 2 points, 5+ days = 3 points (0 points if no vomiting); maximum of 4–5 stools per day = 1 point, 6–7 stools = 2 points, 8+ stools = 3 points (0 points if no diarrhea); presence of fever = 3 points; intravenous fluid received for dehydration = 3 points. This severity score was based on the method described by Lee, *et al*, though we did not collect temperature when fever was reported [19].

Risk factors for AGE included socioeconomic status, nutrition, and secretor and Lewis phenotype. The presence of a toilet in the household at enrollment was a surrogate measure for household income. On a monthly basis, mothers were asked if their child ate any food that is associated with increased AGE risk, including seafood (any kind), raw fruits or vegetables (any kind), or any food outside of the home. Considering the increasing evidence of the importance of host genetic factors in modifying susceptibility to infections, including in a prior study conducted in the same birth cohort [17], we adjusted for the child's secretor and Lewis phenotypes. Due to the additional potential for mother's HBGAs phenotypes to be associated with AGE risk [16], we also adjusted for mother's secretor and Lewis phenotypes.

## Sample collection and laboratory methods

Stool samples were collected from children with AGE within 10 days of the onset of symptoms. Specimens were collected from a plastic container or a soiled diaper retrieved from the

household within two hours of defecation and transported at 4˚C to the Microbiology Department of UNAN-León for analysis. A 10% (wt/vol) suspension of stool was prepared using phosphate-buffered saline (pH = 7.2). The samples were tested for norovirus and sapovirus within 24 hours of collection.

Viral RNA extraction was performed on collected stool samples using the QIAamp Viral RNA Mini Kit (Qiagen, Hilden, Germany), according to manufacturer's instructions. Extracted viral RNA from stool suspensions was analyzed by reverse transcriptase quantitative polymerase chain reaction (RT-qPCR) for norovirus as previously described [20]. Sapovirus infection was detected by RT-qPCR in stools as previously described [7]. Briefly, RT-qPCR was performed with the AgPath-ID OneStep RT-qPCR Kit (Thermo Fisher Scientific, Waltham, MA, USA) using the ABI 7500 Real-Time PCR System. A sample was considered positive for norovirus or sapovirus if the Ct value was $\leq$ 35. HBGA phenotypes were determined by analysis of saliva samples collected from cohort children at 6 months of age. Secretor and Lewis phenotypes were determined using an in-house saliva-based ELISA as described previously [21]. In brief, secretor antigens (α1,2 fucose) present in saliva were recognized using the *Ulex europaeus* lectin peroxidase (UEA-I, Sigma-Aldrich, Sweden) and Lewis antigens using monoclonal anti-Lewis-a and anti-Lewis-b from Seraclone and Diaclone (Bio-Rad, Uppsala, Sweden), respectively. Saliva with optical reading using UEA-I were classified as secretors; and saliva with optical reading using anti-Lewis a or anti-Lewis b were classified as Lewis positive. If the result was indeterminate, we tested the samples collected at 12 or 24 months of age. Saliva samples from mothers were collected at the end of the study and tested for HBGAs phenotypes [22].

## Data analysis

Breastfeeding exclusivity and duration, based on weekly household surveys, and incidence of norovirus or sapovirus AGE episodes were summarized using descriptive statistics. Crude and adjusted hazard ratios (HR) for the association between breastfeeding and norovirus or sapovirus AGE were estimated using time-dependent Cox proportional hazard models. Breastfeeding behavior was measured weekly based on mothers' reports. All children were followed from baseline until 12 months of age, until they experienced a norovirus or sapovirus AGE episode, or until they were lost to follow-up. This analysis is restricted to the first episode reported for each child. Hazard ratios were adjusted for potentially confounding factors, including socioeconomic factors, consumption of high-risk foods, and the child's and mother's secretor and Lewis phenotypes (positive or negative). Data were analyzed using SAS version 9.4 (Cary, North Carolina) and Stata 14 (College Station, Texas).

This study was approved by Institutional Review Boards at the University of North Carolina School of Medicine (IRB #: 20–0698), the National Autonomous University of Nicaragua (UNAN)-León (IRB #: 00003342,) and CDC (Research determination: 0900f3eb81c526a7).

## Results

### Study population characteristics

Of 444 children recruited in the study, 443 (99.8%) completed at least one weekly follow-up visit and were included in the analysis. By the 12-month visit, 375 children (84.6%) remained enrolled in the study and 68 had withdrawn. Mothers of all 443 children reported having breastfed their children at least once in the first 12 months of life. Half of the children received exclusive breastfeeding at some point during the study (Table 1). The median duration of exclusive breastfeeding was 2 weeks (IQR: 1–4 weeks), and the median duration of any breastfeeding was 43 weeks (IQR: 18–49 weeks) (Table 1). Sixty-nine children (15.6%) experienced

**Table 1. Characteristics associated with risk of acute gastroenteritis among children in a Nicaraguan birth cohort (n = 443).**

| Variable | n (%) or median (Interquartile Range) |
| --- | --- |
| Sex | |
| Male | 226 (51.0) |
| Female | 217 (48.9) |
| Baseline risk factors | |
| Presence of a toilet in the household | 323 (72.9) |
| Child secretor-positive (n = 443) | 395 (89.4) |
| Mother secretor-positive (n = 312) | 244 (78.5) |
| Child Lewis-positive (n = 443) | 373 (84.2) |
| Mother Lewis-positive (n = 312) | 225 (72.4) |
| Breastfeeding | |
| Number of children who received exclusive breastfeeding for any duration | 224 (50.6) |
| Median weeks of exclusive breastfeeding (IQR) | 2 (1–4) |
| Median weeks of any breastfeeding (IQR) | 43 (18–49) |
| Norovirus and sapovirus episodes | |
| Sapovirus AGE episodes in first 12 months of life | 34 (7.7) |
| Median age of child at first sapovirus AGE episode | 8.4 (6.5–9.4) |
| Norovirus AGE episodes in first 12 months of life | 69 (15.6) |
| Median age of child at first norovirus AGE episode | 8.1 (5.7, 9.9) |

an AGE episode associated with norovirus genogroups GI and GII, and 34 children (7.8%) experienced a sapovirus AGE episode in the first year of life. The median age at the first norovirus or sapovirus infection was approximately 8 months (Fig 1). The first peak of norovirus infections occurred in January 2018, with subsequent peaks in mid- and late-2018 (Fig 2). A single peak was identified for sapovirus, in mid-2018.

## Severity of norovirus and sapovirus infections

Norovirus and sapovirus episodes were of moderate severity, having severity scores of 5 and 4.5, respectively, on a scale of 0–15. Four of the 69 norovirus episodes (5.8%) and three of the 34 sapovirus episodes were associated with diarrhea lasting more than 14 days, thereby becoming persistent gastroenteritis [23]. Only norovirus episodes were associated with hospitalization (n = 4), three of which required intravenous rehydration.

## Maternal and child histo-blood group antigen profiles

Secretor phenotype was available for 311 mothers, 244 (78.5%) of whom were secretors, and for 442 children, 395 (89.4%) of whom were secretors (Table 2). Most norovirus episodes occurred among secretor-positive (98.5%) and Lewis-positive (85.5%) children, as did most of the 34 sapovirus episodes (88.2% each for secretor and Lewis positive) (Table 2). Approximately three-quarters of norovirus and sapovirus cases occurred among children with secretor-positive or Lewis-positive mothers. For norovirus, we observed inverse associations between secretor-positive and secretor-negative children in risk of norovirus AGE; this association was less pronounced with respect to maternal secretor status, and no such associations were observed for sapovirus (Table 2).

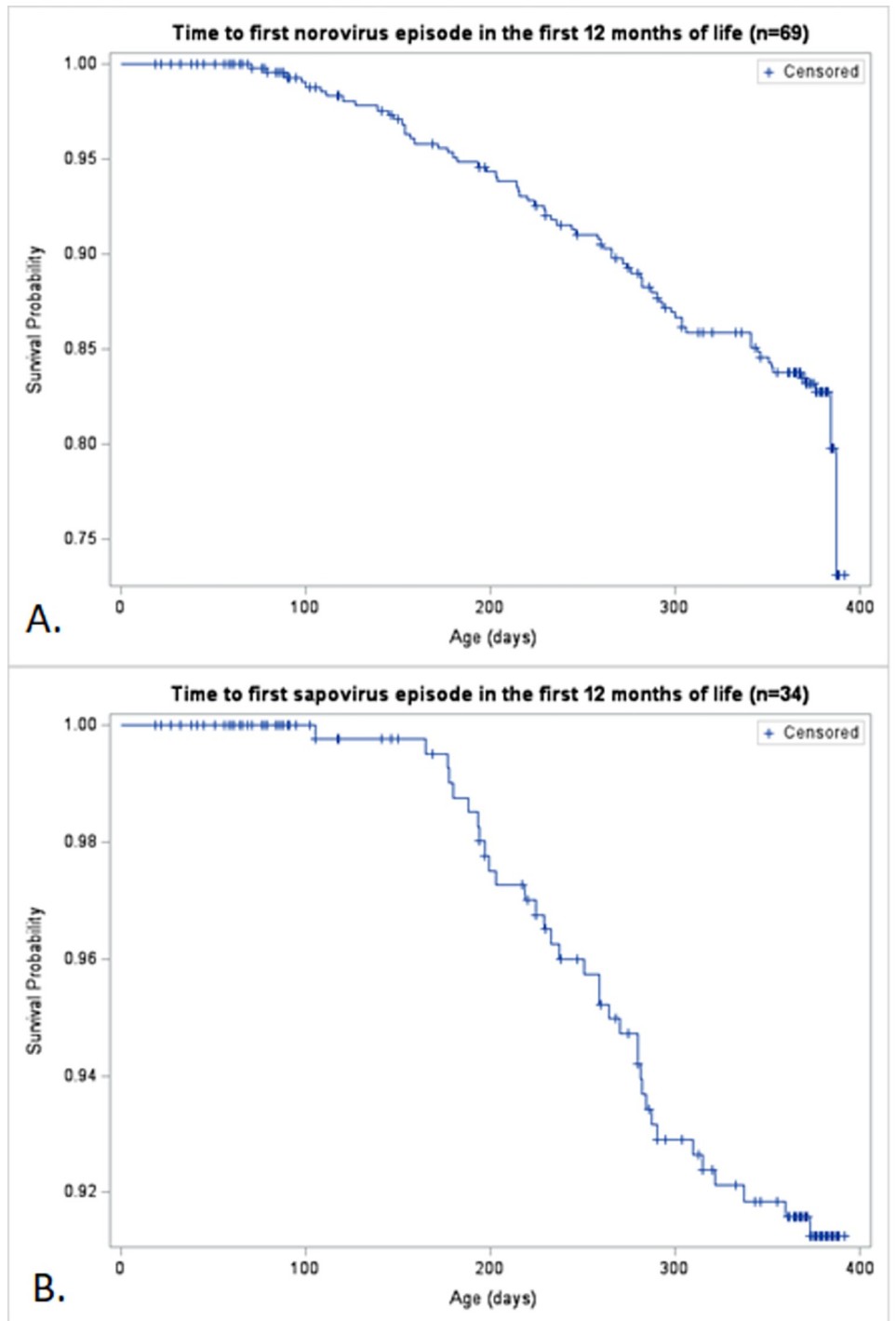

**Fig 1. Time to first norovirus and sapovirus AGE episode in the first 12 months of life.** This figure illustrates the Kaplan-Meier survival curve for the 69 first norovirus episodes (Panel A), and the 34 first sapovirus episodes (Panel B).

## Relative hazard of norovirus or sapovirus AGE by breastfeeding frequency

The model adjusted for socioeconomic factors, consumption of high-risk foods, and child's and mother's genetic phenotypes showed no protective association of breastfeeding against

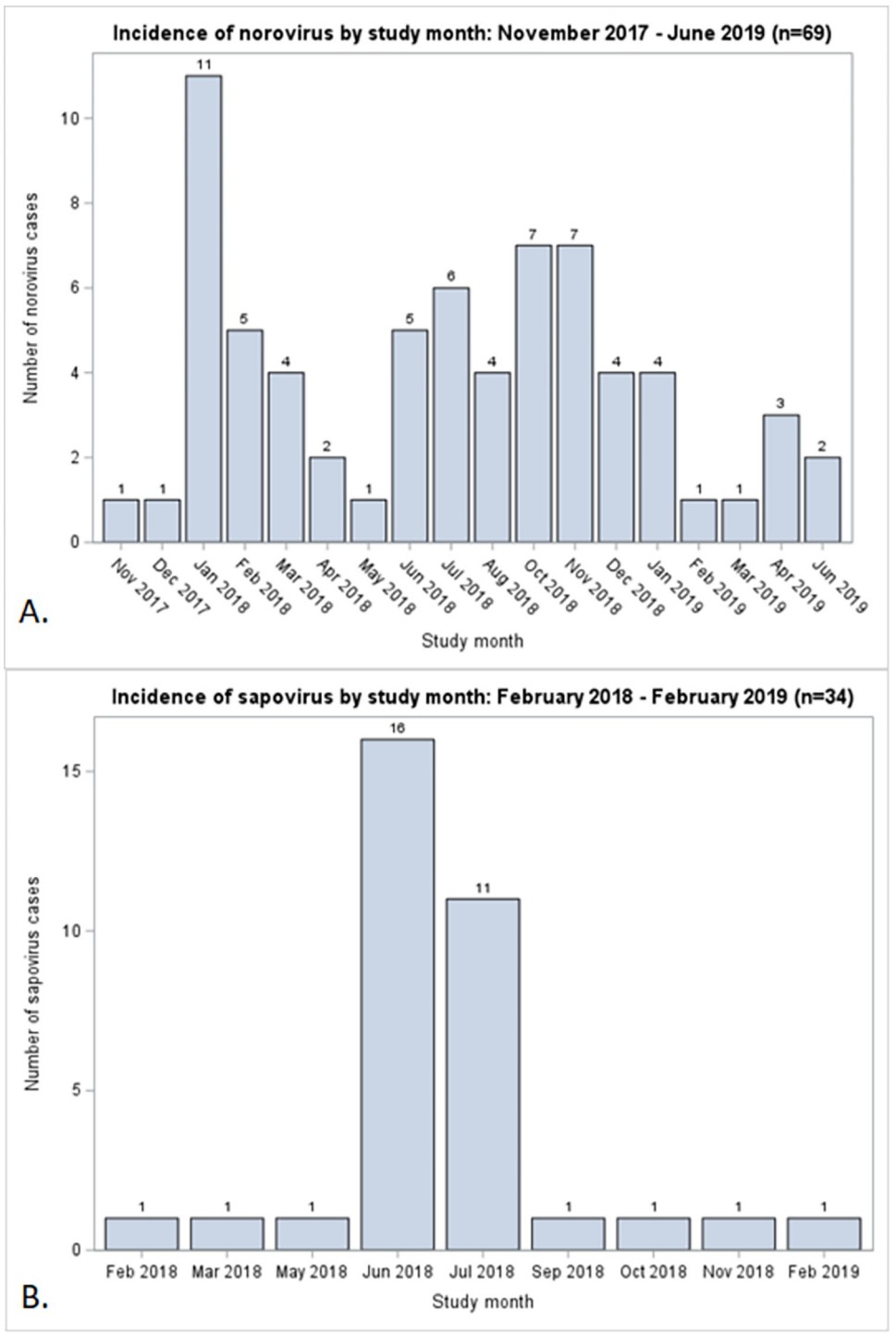

**Fig 2. Seasonality of norovirus and sapovirus AGE episodes in the first 12 months of life.** This figure illustrates the timing of the 69 first norovirus episodes (Panel A) and the 34 first sapovirus episodes (Panel B) by month and year.

norovirus episodes (HR: 1.09, 95% CI: 0.62, 1.92) (Table 3). In this model, child secretor-positive phenotype was the strongest risk factor for norovirus AGE (HR: 11.02, 95% CI: 1.50, 80.80), whereas mother secretor-positive phenotype was the strongest protective factor for

**Table 2. Child and maternal host genetic factors and incidence of norovirus and sapovirus.**

| | Norovirus + | Norovirus - | Sapovirus + | Sapovirus - |
|---|---|---|---|---|
| *Child (n = 442)* | *n = 69 (%)* | *n = 372 (%)* | *n = 34 (%)* | *n = 408 (%)* |
| Secretor + (n = 394; 1 missing) | 68 (98.5) | 326 (87.6) | 30 (88.2) | 365 (89.4) |
| Secretor—(n = 47) | 1 (1.4) | 46 (12.3) | 4 (11.7) | 43 (10.5) |
| Lewis + (n = 373) | 59 (85.5) | 313 (84.1) | 30 (88.2) | 343 (84.0) |
| Lewis—(n = 70) | 10 (14.4) | 60 (16.1) | 6 (1.8) | 66 (16.1) |
| *Mother (n = 311)* | *n = 61* | *n = 250* | *n = 25* | *n = 286* |
| Secretor + (n = 244) | 46 (75.4) | 198 (79.2) | 18 (72.0) | 226 (79.0) |
| Secretor—(n = 67) | 15 (24.5) | 52 (20.8) | 7 (28.0) | 60 (20.9) |
| Lewis + (n = 225) | 45 (73.7) | 180 (72.0) | 19 (76.0) | 206 (72.0) |
| Lewis—(n = 86) | 16 (26.2) | 70 (28.0) | 6 (24.0) | 80 (27.9) |

norovirus AGE (HR: 0.55, 95% CI: 0.29, 1.05). Lewis-positive phenotype of the child or the mother was not significantly associated with protection against norovirus AGE. For sapovirus, adjusted estimates showed no protective association of breastfeeding (HR: 1.00, 95% CI: 0.82, 1.21). For both norovirus and sapovirus, having a toilet in the household tended to confer some protection against clinical disease, as did consumption of high-risk foods (seafood, raw fruits or vegetables) and food outside the household (Table 3). For both pathogens, interaction terms between nutrition and socioeconomic factors were not statistically significant.

Our study sample had a higher proportion of children who were secretor-positive compared to mothers. When the analysis was restricted to mothers of secretor-positive children, the protective association between mother secretor-positive phenotype showed slightly greater protection against norovirus AGE (HR: 0.52, 95% CI: 0.27, 1.00).

In a post-hoc analysis, we estimated the association between breastfeeding and severity of AGE episodes among children who experienced symptomatic norovirus (n = 69) and sapovirus (n = 34) infections. We used the median severity score (norovirus: 5; sapovirus: 4.5) as the cutoff for severe versus non-severe episodes. Because of the reduced number of outcomes in this analysis, the HR estimates were highly imprecise and did not suggest a definitive

**Table 3. Crude and adjusted relative hazards between breastfeeding and norovirus and sapovirus AGE in first 12 months of life.**

| | Norovirus AGE (n = 69) | | Sapovirus AGE (n = 34) | |
|---|---|---|---|---|
| | Crude HR | Adjusted HR[1] | Crude HR | Adjusted HR[1] |
| Breastfeeding in the last week | | | | |
| Any vs. none | 1.31 (0.78, 2.21) | 1.09 (0.62, 1.92) | 0.99 (0.84, 1.17) | 1.00 (0.82, 1.21) |
| Household sanitation | | | | |
| Toilet vs. latrine or no sanitation | 0.56 (0.35, 0.91) | 0.66 (0.38, 1.12) | 0.58 (0.29, 1.16) | 0.46 (0.21, 1.02) |
| Consumption of high-risk foods in the last month | | | | |
| Any vs. none | 0.53 (0.29, 0.98) | 0.69 (0.34, 1.41) | 0.35 (0.16, 0.78) | 0.36 (0.14, 0.90) |
| Child's HBGA profile | | | | |
| Secretor+ vs. Secretor- | 9.60 (1.33, 69.15) | 11.02 (1.50, 80.80) | 0.92 (0.32, 2.60) | 1.71 (0.39, 7.51) |
| Lewis+ vs Lewis- | 0.97 (0.49, 1.89) | 0.88 (0.40, 1.94) | 1.28 (0.45, 3.62) | 1.63 (0.36, 7.27) |
| Mother's HBGA profile | | | | |
| Secretor+ vs. Secretor- | 0.81 (0.45, 1.44) | 0.55 (0.29, 1.05) | 0.68 (0.28, 1.62) | 0.46 (0.36, 7.27) |
| Lewis+ vs Lewis- | 1.11 (0.63, 1.97) | 1.27 (0.66, 2.46) | 1.22 (0.49, 3.06) | 1.60 (0.55, 4.68) |

HR = Hazard ratio

[1]Hazard ratios adjusted for all other variables in the table.

association between breastfeeding and AGE severity for norovirus (adjusted HR: 1.56, 95% CI: 0.65, 3.73) or sapovirus (adjusted HR: 1.46, 95% HR: 0.27, 7.85).

## Discussion

Breastfeeding was predominantly non-exclusive among the mother-infant pairs in this Nicaraguan birth cohort and was not associated with protection against norovirus and sapovirus AGE episodes. Child's secretor-positive phenotype was a strong risk factor for norovirus AGE [17], whereas mother's secretor-positive phenotype tended to be protective against norovirus AGE in their children. These influences of secretor status were not observed for sapovirus AGE, consistent with findings from prior studies indicating that sapovirus infects individuals of all type I HBGA phenotypes [18, 24, 25]. A previous study further showed no binding of sapovirus VLPs to type I HGBAs, corroborating the lack of association between secretor or Lewis status on sapovirus infection risk [26].

The most recent World Health Organization (WHO) guidelines for infant and young child feeding recommend exclusive breastfeeding for the first six months of life and list protection against gastrointestinal infections among one of the chief benefits of exclusive breastfeeding [27]. Most mothers in the cohort (98%) reported ever breastfeeding at enrollment. However, the duration of exclusive breastfeeding in our study was short, which prevented us from arriving at definitive conclusions regarding the association between exclusive breastfeeding and norovirus or sapovirus AGE. According to UNICEF, many Latin American countries, including Nicaragua, have suboptimal breastfeeding practices, with only 32% of children younger than six months being exclusively breastfed [28]. A prior study of breastfeeding practices in León found only 12.7% exclusive breastfeeding for six months, and indicated that travel time to a health center greater than one hour, as a proxy measure for living in a remote or rural area, was associated with exclusive breastfeeding in the first six months of life [29]. In León, most women have access to a health post in their neighborhoods, suggesting that exclusive breastfeeding is less common in urban-dwelling women, and might be associated with greater access to infant formulas and other feeding options.

Observed seasonality of norovirus infections are consistent with prior studies from León [12, 30], and ours is the first study to show high regional sapovirus incidence in June and July, consistent with the rainy season. Children experienced their first norovirus or sapovirus episodes at approximately 8 months of age, long after most mothers had stopped breastfeeding exclusively and began introducing supplementary foods, suggesting that some maternally-acquired immunity prevented infections in early infancy. However, Nicaraguan women are guaranteed eight weeks of paid maternity leave after delivery, which may preclude exclusive breastfeeding for the recommended six months as mothers return to the workforce [31]. Cultural and social factors may also impact breastfeeding practices among young mothers in urban areas, as they are uncomfortable breastfeeding in public or believe that prolonged breastfeeding will negatively affect breast shape or volume [32, 33]. Given the limited duration of exclusive breastfeeding in our population (median of 2 weeks), future studies should be conducted in a different population where exclusive breastfeeding is practiced more widely and for longer duration, or as part of a clinical trial that provided lactation support personnel who would constantly advise and encourage new mothers enrolled in the study to continue exclusive breastfeeding. These studies would allow us to better understand the association between norovirus or sapovirus AGE and exclusive breastfeeding.

We were also surprised that consumption of high-risk foods and eating food outside of the home were associated with protection against sapovirus AGE. Sapovirus infections are usually transmitted by the fecal-oral route, and are less likely to be caused by foodborne outbreaks

compared to norovirus [34]. This might explain why the risk of sapovirus AGE was not affected by adjusting for consumption of high-risk foods, whereas protective association with norovirus was attenuated and nullified on adjustment. We found that consumption of seafood and fresh fruits and vegetables was positively associated with having piped water in the home, which is a marker of higher socioeconomic status. If consumption of high-risk foods or other nutritional practices were associated with other factors that were protective against sapovirus AGE, the relationship between high-risk foods and sapovirus AGE could be confounded.

Our study, like others, shows that a higher risk of norovirus AGE in the first year of life is associated with child's secretor-positive phenotype but not with Lewis-positive phenotype [16, 17, 35]. However, an important finding in our study is the protective association of mother's secretor-positive phenotype on norovirus AGE, unlike other studies which have found fewer norovirus infections among children of non-secretor mothers [36], an enhanced protection against all-cause diarrhea in breastfed children of non-secretor mothers [37], or no effect on norovirus infections [16]. The protective effect of mother's secretor-positive phenotype noted in our study might be explained by higher concentrations of specific human milk oligosaccharides (HMO) and HBGAs in breastmilk, which in turn can inhibit norovirus binding to the infant's intestinal epithelium [38–40]. It is also possible that breastmilk from secretor-positive mothers contains higher titers of norovirus-specific IgA that can neutralize norovirus in the infant's gut [36, 41, 42]. Another possible explanation is that secretor-positive mothers might transfer higher titers of norovirus-specific IgG antibodies with broader specificity to their children through the placenta, and such maternal antibodies contribute to limited infections very early in life; the protective effect of maternal antibodies on norovirus disease has not been extensively explored [43]. Future studies should further investigate these associations to better understand the role of maternal secretor status on risk of norovirus or sapovirus AGE.

A limitation of this study was the limited duration of exclusive breastfeeding, which precluded making valid inferences regarding the association between exclusive breastfeeding and norovirus or sapovirus AGE. Furthermore, 30% of mothers had missing data for the secretor and Lewis phenotypes, and we did not impute missing data for this covariate. Thus, these mothers were excluded from the adjusted analyses and the statistical power was decreased. Studies of breastfeeding behavior may be subject to social desirability bias, as mothers may be more likely to report exclusive breastfeeding in accordance with Ministry of Health recommendations; however, the low reported frequency of exclusive breastfeeding suggests that this bias plays a minor role in our study. We did not assess routinely-collected stools for asymptomatic calicivirus infections, nor did we test all AGE stools for a wide panel of enteric infections. Thus, we cannot conclude that symptoms were caused by norovirus or sapovirus infections specifically. It is possible that breastfeeding is associated with protection against other enteric pathogens that we did not assess in the current analysis. Ongoing studies are assessing pathogen-specific associations with breastfeeding across a wider array of pathogens, accounting for the composition of breastmilk with respect to IgA antibodies and HMOs as potential protective factors against AGE.

Despite these limitations, our study is one of the few studies to focus on breastfeeding as a primary exposure to understand its association with norovirus or sapovirus AGE. In contrast to previous studies that explored breastfeeding as a fixed variable, our study provides more granular insight into breastfeeding treated as a weekly varying exposure, an approach that closely mirrors actual practices. Our findings reinforce the importance of continuing to study the protective benefits of breastfeeding and the mechanisms by which breastfeeding can prevent early childhood infections. Our results also suggest that breastfeeding, especially non-exclusive breastfeeding, may not be sufficient to reduce AGE associated with all enteric pathogens, and that combinations of interventions to reduce childhood AGE are warranted.

## Supporting information

**S1 Dataset.**
(XLSX)

**S2 Dataset.**
(XLS)

**S3 Dataset.**
(XLS)

## Acknowledgments

We greatly appreciate the participation of the parents of the children contributing to the SAGE cohort, and recognize the immense efforts of the laboratory and fieldwork team: Merling Balmaceda, Vanessa Bolaños, Nancy Corea, Jhosselyng Delgado, Marvel Fuentes, Yadira Hernández, Llurvin Madríz, Patricia Mendez, Yuvielka Martínez, María Mendoza, Ruth Neira, Xiomara Obando, Verónica Pravia, Yorling Picado, Aura Scott, and Mileydis Soto. The authors are grateful for support from the local Ministry of Health office in León, and we thank the personnel from the Perla María Norori Health Center for their administrative support in recruitment of study participants. The findings and conclusions in this article are those of the authors and do not necessarily represent the official position of the Centers for Disease Control and Prevention.

## Author Contributions

**Conceptualization:** Ruthly François, Emilya Huseynova, Fredman González, Samuel Vilchez, Jan Vinjé, Sylvia Becker-Dreps, Filemon Bucardo.

**Data curation:** Fredman González.

**Formal analysis:** Nadja Alexandra Vielot, Emilya Huseynova, Fredman González, Yaoska Reyes, Lester Gutierrez.

**Funding acquisition:** Sylvia Becker-Dreps, Filemon Bucardo.

**Investigation:** Nadja Alexandra Vielot, Fredman González, Yaoska Reyes, Lester Gutierrez, Christian Toval-Ruiz, Samuel Vilchez, Sylvia Becker-Dreps, Filemon Bucardo.

**Methodology:** Johan Nordgren, Samuel Vilchez, Jan Vinjé, Sylvia Becker-Dreps.

**Project administration:** Samuel Vilchez, Sylvia Becker-Dreps, Filemon Bucardo.

**Resources:** Nadja Alexandra Vielot, Samuel Vilchez, Jan Vinjé, Sylvia Becker-Dreps, Filemon Bucardo.

**Software:** Nadja Alexandra Vielot.

**Supervision:** Nadja Alexandra Vielot, Samuel Vilchez, Sylvia Becker-Dreps, Filemon Bucardo.

**Writing – original draft:** Nadja Alexandra Vielot, Ruthly François, Emilya Huseynova, Sylvia Becker-Dreps.

**Writing – review & editing:** Nadja Alexandra Vielot, Ruthly François, Emilya Huseynova, Fredman González, Yaoska Reyes, Johan Nordgren, Christian Toval-Ruiz, Samuel Vilchez, Jan Vinjé, Sylvia Becker-Dreps, Filemon Bucardo.

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
