## [Decision Letter · Decision Letter 0]

29 Aug 2022

PONE-D-22-10820Association between breastfeeding, host genetic factors, and calicivirus gastroenteritis in a Nicaraguan birth cohortPLOS ONE

Dear Dr. Vielot,

Thank you for submitting your manuscript to PLOS ONE. After careful consideration, we feel that it has merit but does not fully meet PLOS ONE’s publication criteria as it currently stands. Therefore, we invite you to submit a revised version of the manuscript that addresses the points raised during the review process.

This manuscript was evaluated by three reviewers whose comments are included below. Overall, the reviewers provided positive feedback but requested revisions to address points pertaining to reporting and aspects of the data analyses and results. Please address all of the reviewers' comments through revisions to your manuscript.

We look forward to receiving your revised manuscript.

Kind regards,

Renee Hoch, Ph.D.

Managing Editor, PLOS Publication Ethics

PLOS ONE

Journal Requirements:

"This study was supported by award R01AI127845 from the National Institute of Allergy and Infectious Diseases (NIAID). SBD is supported by K24AI141744 from NIAID; FG, YR, and LG are supported by an international research capacity-building award from the NIH-Fogarty International Center, D43TW010923. "

Reviewers' comments:

Reviewer's Responses to Questions

**Comments to the Author**

1. Is the manuscript technically sound, and do the data support the conclusions?

Reviewer #1: Yes

Reviewer #2: Yes

Reviewer #3: Yes

2. Has the statistical analysis been performed appropriately and rigorously? 

Reviewer #1: Yes

Reviewer #2: Yes

Reviewer #3: Yes

3. Have the authors made all data underlying the findings in their manuscript fully available?

Reviewer #1: No

Reviewer #2: No

Reviewer #3: No

4. Is the manuscript presented in an intelligible fashion and written in standard English?

Reviewer #1: Yes

Reviewer #2: Yes

Reviewer #3: Yes

5. Review Comments to the Author

Reviewer #1: PONE-D-22-10820

This study provides important information on associations between breastfeeding and acute gastroenteritis (AGE) caused by norovirus and sapovirus among children in a birth cohort in Leon, Nicaragua. This study has very rich weekly data on breastfeeding and AGE and monthly data on other risk factors, enabling the authors to conduct a time-varying Cox hazard analysis. Results are interesting. Some of them align with previous findings and others don’t, and the authors provide good discussion on each point, which was helpful. I have a few comments.

Results

- Regarding the protective effect of the consumption of high-risk foods and eating food outside, I agree that it is surprising. The authors discuss potential reasons behind this in the Discussion, which was helpful. Could you also include an interaction term between the variable for the comsumption of high-risk foods and the variable for the socioeconomic status in your Cox hazard model? That might be helpful in interpreting the relationship.

Discussion

- I do not know any background in Nicaragua, and did not understand why exclusive breastfeeding was associated with travel time to a health center. What does a health center/post do with breastfeeding?

- Are there any studies or articles you could cite about cultural and social factors associated with short-term exclusive breastfeeding that you talked about in LIne 236-238 (e.g., prolonged breastfeeding negatively affecting breast shape and volume)?

- I guess a natural question from readers would be “so what about other pathogens tested in this study?” but I guess the authors have a plan to write another report on that.

Reviewer #2: In this study, Vielot and coauthors analyzed the association between breastfeeding practices, child/mother genetic factors, and norovirus/sapovirus AGE using a birth cohort data obtained in Nicaragua. Authors found strong associations between child secretor status and norovirus infections in their first year of life but authors could not find conclusive evidence of associations between breastfeeding and protection against AGE caused by norovirus and sapovirus due to a limited sample size and the fact that majority of mothers performed exclusive breastfeeding for only a few weeks.

While this study could not provide conclusive evidence of associations between breastfeeding and viral AGE, findings provided in this study are important to further conduct and understand the roles of exclusive and non-exclusive breastfeeding on norovirus/sapovirus AGE. Thus, it implies that non-exclusive breastfeeding may not protect children from norovirus/sapovirus AGE, and warrants further studies with different population where exclusive breastfeeding is performed commonly for a long duration or under the lactation support. Please find below the comments/suggestions to improve the manuscript before publication.

1. Authors claimed that "most norovirus episodes occurred among secretor positive and Lewis positive children...", but it could be confusing and mislead an incorrect conclusion as most of the children enrolled and studied in this cohort were secretor positive and Lewis positive. In fact, authors analyzed the HR with host genetic factors in Table 3 to show that there is no association between Lewis profiles and norovirus/sapovirus AGE. I would suggest to rephrase the section "Maternal and child HBGA profiles" or omit or combine this section with the next section.

2. One of the interesting findings in this study is that they did not find any associations between host genetic factors (neither secretor nor Lewis) and sapovirus AGE. Is there any other studies that indicate the roles (or no roles) of host genetic factors on sapovirus infections? If so please consider to include findings from such previous studies and provide more discussions on the roles of secretor status on sapovirus infections.

3. Another interesting observation is that consumption of high-risk foods was protective against sapovirus AGE. Authors suggested that it could be due to confounding factors such as nutritional practices or socioeconomic status, and I agree with authors but please also consider the epidemiological observations that foodborne ourbreaks are less common in sapovirus as compared to norovirus. I believe there is no conclusive evidence on the risk of food consumption on sapovirus AGE, but if the risk is lower in sapovirus than in norovirus, it may explain the differences of the observation between norovirus AGE and sapovirus AGE in this study. Thus, it showed significant protective associations only among sapovirus AGE but not in norovirus AGE as food consumption could work as a risk factor only (or more) in norovirus AGE.

4. I understand typing of host genetic factors was performed in their previous study, but please consider to describe again in this manuscript how the host genetic factors were determined in the method section.

Reviewer #3: Violet et al studied the association between calicivirus AGE and exclusive breastfeeding or host HBGA status in a Nicaragua birth cohort from June 2017 to July 2018. This study was very well planned out and its findings on how maternal and child secretor status influences norovirus disease risk is not unexpected, nevertheless very interesting. This manuscript is very well written.

Minor questions and comments:

Were there seasonal trends in sapovirus and norovirus positivities? Were they consistent with what's typically seen in the region?

Line 181-182, diarrhea lasting more than 7 days is not considered as acute gastroenteritis in most clinical definitions. Consider reword/or update definition of AGE in methods.

Line 201-202, If secretor positive were a dominate trait, it would means 3/4 progeny would have the trait from AaXAa heterozygous parents, all progeny if one parent is AA and 1/2 if from AaXaa parents. I wouldn't call it surprising to see more secretor positive children than mothers. Also, it is a bit more complicated than a simple Mendelian hereditary pattern since there are weak secretor alleles that are semi-functional. I would suggest not to get into the genetics of HBGA here.

6. PLOS authors have the option to publish the peer review history of their article (what does this mean?). If published, this will include your full peer review and any attached files.

Reviewer #1: No

Reviewer #2: No

Reviewer #3: No

---

## [Author Response · Author response to Decision Letter 0]

9 Sep 2022

Please see the attached "Response to reviewers" document for a point-by-point response to each reviewer comment.

---

## [Editor Report · Decision Letter 1]

19 Sep 2022

Association between breastfeeding, host genetic factors, and calicivirus gastroenteritis in a Nicaraguan birth cohort

PONE-D-22-10820R1

Dear Dr. Vielot,

We’re pleased to inform you that your manuscript has been judged scientifically suitable for publication and will be formally accepted for publication once it meets all outstanding technical requirements. Please let me disclose that I participated as a reviewer for the initial evaluation of this manuscript, and now I am working as an Academic Editor to make this decision. As an editor, I have a very minor comment and I would suggest to clarify the terms, socioeconomic status and nutrition, in the manuscript. I believe you used the presence of a toilet (a surrogate measure of house income) as socioeconomic status and consumption of specific food as nutrition, but it is not clearly explained in the method section. 

Kind regards,

Kentaro Tohma

Guest Editor

PLOS ONE
---

## [Editor Report · Acceptance letter]

5 Oct 2022

PONE-D-22-10820R1 

Association between breastfeeding, host genetic factors, and calicivirus gastroenteritis in a Nicaraguan birth cohort 

Dear Dr. Vielot:

I'm pleased to inform you that your manuscript has been deemed suitable for publication in PLOS ONE. Congratulations! Your manuscript is now with our production department. 

Kind regards, 

on behalf of

Dr. Kentaro Tohma 

Guest Editor

PLOS ONE